# The Influence of Liquid Low-Radioactive Waste Repositories on the Mineral Composition of Surrounding Soils

**Victoria Krupskaya** [1,2,*], **Anatoliy Boguslavskiy** [3], **Sergey Zakusin** [1,4], **Olga Shemelina** [3], **Mikhail Chernov** [4], **Olga Dorzhieva** [1] **and Ivan Morozov** [1]

1   Institute of Ore Geology, Petrography, Mineralogy and Geochemistry, Russian Academy of Science (IGEM RAS), 119017 Moscow, Russia; zakusinsergey@gmail.com (S.Z.); dorzhievaov@gmail.com (O.D.); ivan.morozov@yandex.ru (I.M.)
2   Nuclear Safety Institute, Russian Academy of Science (IBRAE RAS), 115191 Moscow, Russia
3   V.S. Sobolev Institute of Geology and Mineralogy Siberian Branch Russian Academy of Sciences (IGM SB RAS), 630090 Novosibirsk, Russia; boguslav@igm.nsc.ru (A.B.); shem@igm.nsc.ru (O.S.)
4   Geological Faculty, M.V. Lomonosov Moscow State University, 119991 Moscow, Russia; miha.chernov@yandex.ru
*   Correspondence: krupskaya@ruclay.com

**Abstract:** Clay minerals may transform in various systems under the influence of geological, biological, or technogenic processes. The most active to the geological environment are technogenic and biochemical processes that, in a relatively short time, can cause transformation of the rocks' composition and structure and formation of new minerals, especially clay minerals. Isolation of radioactive waste is a complex technological problem. This work considers the influence of alkaline solutions involved in the radioactive waste (RW) disposal process. In the Russian Federation, due to historical reasons, radioactive waste has accumulated in various types of repositories and temporary storages. All these facilities are included in the federal decommissioning program. Solid radioactive wastes in cement slurries at the landfill site of the Angara Electrolysis Chemical Combine are buried in sandstones and currently suffer the influence of a highly alkaline and highly saline groundwater storage area, which leads to a considerable transformation of the sandstones. This influence results in the formation of peculiar "technogenic" illites that have smectite morphology but illite structure which was confirmed by modeling of X-ray diffraction (XRD) patterns. The described transformations will lead to the increase of porosity and permeability of the sandstones. The research results can be used in assessing the potential contamination of the areas adjacent to the disposal site and in planning the decommissioning measures of this facility.

**Keywords:** low-radioactive waste; waste facilities; highly concentrated solutions; clay minerals transformations

## 1. Introduction

The problem of radioactive waste (RW) disposal at nuclear facilities is very relevant. The main goal in this process is maintaining radiation and toxic safety during the whole lifetime of the repository [1]. At present, in Russia, there are various nuclear and radiation hazardous facilities of nuclear legacy requiring decommissioning activities. They are included in the federal program for decommissioning [2,3].

During the initial stages of the development of the nuclear industry, waste solutions with low pH were neutralized by calcium hydroxide. Then, the highly concentrated solutions with residual

radionuclides were drained in subsurface ponds—sludge repositories [3]. These solutions interacted with underlying soils, changing their structural, chemical, and mineralogical composition [4].

The fate of repositories is a key problem after decommissioning. If the existing barriers are efficient enough to isolate radioactive wastes, underlying and host soils have low permeability together with high sorption characteristics, RW recovery and retrieval can carry serious environmental risks compared to the in situ preservation. If there is a risk of penetration of the contaminants in aquifers, either storage modernization or RW retrieval is needed [5]. An environmental impact assessment and study of the forms of migration and the state of barriers preventing the spread of contaminants outside the repositories should be conducted to understand the sustainability of existing facilities [6].

The best material for subsurface storage preservation is argillaceous sediments, which both prevent the groundwater flow and precipitate the dissolved radionuclides. Clay minerals form in different systems under geological, biological, or technological processes. Industrial and biochemical processes are believed to be the most sensitive towards the host geological environment. For a short period, they lead to the transformation of composition and structure of rocks and cause mineral, primarily clay, neoformations to appear [7,8].

Many studies of changes in the geological environment in the vicinity of low- and medium-level radioactive waste disposal sites are mainly aimed at the migration of radionuclides [9] and the spread of contamination prior to decommissioning operations [10,11]. Works that touch upon the issues of alteration of the mineral composition of rocks in which RW was isolated or which are located in the vicinity to the disposal sites are quite rare [12]. Thus, in works with the authors of this research, the transformations of the composition of clay minerals in sands were studied.

This research aims to determine possible changes in the composition and structure of clay minerals and to predict changes in the filtration properties of soils containing radioactive waste.

## 2. Geological Position and Characteristics of Uranium Recovery Facilities of AECC

The Angarsk Electrolysis Chemical Combine (AECC) is one of the oldest uranium recovery facilities. It is located in the Angarsk, Irkutsk region, Russia. The first output from this plant was issued in 1960. Until 2014, AECC had two interrelated production lines: (1) sublimation unit (production of fluorine and anhydrous hydrogen fluoride, uranium conversion, and transformation into uranium hexafluoride—UHF) and (2) separation unit (separation of uranium isotopes in multistage gas centrifuge cascades to increase the concentration of $^{235}$U isotope in UHF). The sublimate line was stopped in 2014. At present, AECC receives natural raw uranium material in the form of uranium oxide $U_3O_8$ and tetrafluoride $UF_4$ with $^{235}$U content about 0.7%. After enrichment, its concentration increases to 3.5%.

The low-level radioactive waste storage from the AECC is located several kilometers away from the Angarsk city border (Figure 1). It is designed for accumulation and precipitation of limewater suspension produced during the neutralization of liquid nitric acid waste from a chemical plant with $Ca(OH)_2$. The facility consists of six near-surface open reservoirs with sizes of $100 \times 70$ m and $85 \times 90$ m and 17,000 m$^3$ (reservoirs I-IV) and 18,000 m$^3$ (reservoirs V-VI) in volumes. Reservoirs I and II are filled up to the designed level and covered with a clay liner for the prevention of rainfall and melting snow infiltration. Reservoir III is at the stage of conservation and reservoirs IV-VI are still under operation [13].

At the time these repositories were designed, the technological scheme of the AECC was intended to cause precipitation of neutralized suspension and subsequent discharge of the clarified part into the Angara River. Infiltration through the reservoir bottom and walls was not considered. In the late 1980s, the production technology had been changed and the incoming amount of liquid waste sharply decreased. Nowadays, the suspension is separated into a solid precipitate and highly saline supernatant solution. Despite the waterproof measures of the storage, fluids migrate into the subsoils. As a result, a significant change in the groundwater composition is observed around the storage facilities.

The first aquifer is represented by the waters of Quaternary sediments, and the most saturated within the terraces above the floodplain. Depending on the topography, the aquifer lies at depths of 0.5 to 7.0 m. The water of the Quaternary aquifer is bicarbonate Ca > Mg > Na. Total mineralization ranges from 0.15 to 0.3 g/L. In the sampled area, the top of the technogenic altered groundwater flow occurs at a depth of 2.5 to 6.3 m. The groundwater from the solid radioactive waste (SRW) SRW construction site moves in the north–northeast direction towards the Angara river, which is about 5.5 km in the northeast direction. Technogenic waters have a carbonate–nitrate composition (Na > Ca > Mg). Directly under the waste ponds, salinity in some wells reaches 9 g/L (C70), but already at a distance of 300–500 m (C78, 79), it drops to 0.3–0.5 g/L [14].

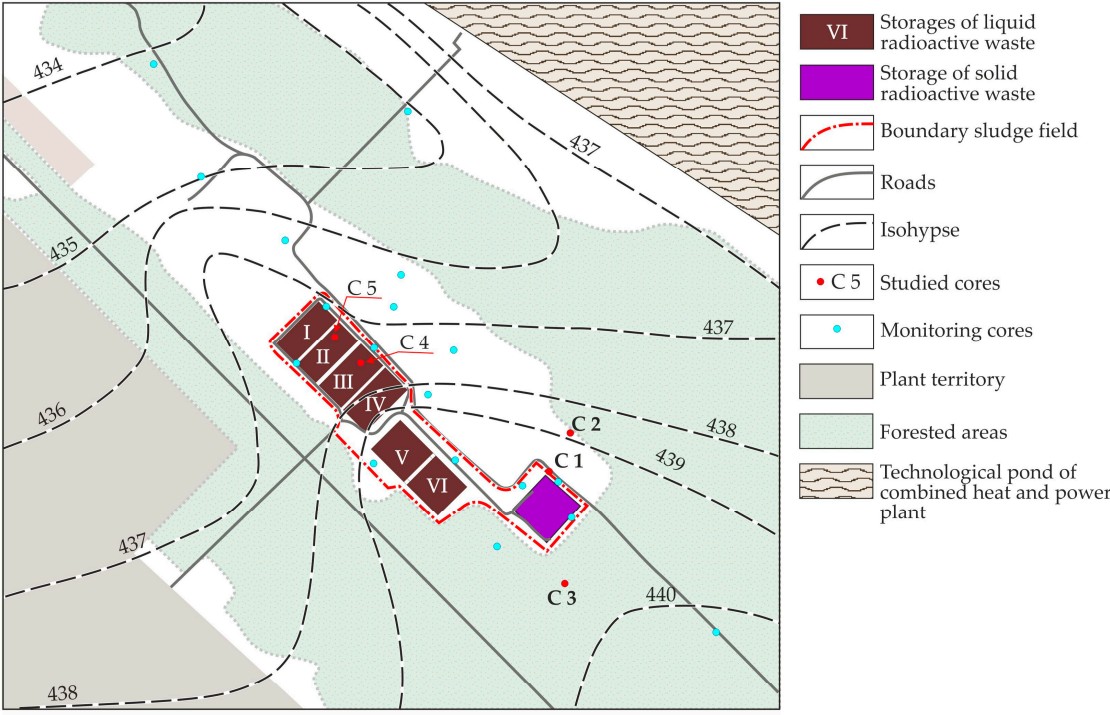

**Figure 1.** Overview scheme of the storage locations.

According to the research from the Irkutsk geological and ecological survey [15], at the moment of two production lines operation, the salinity of waste was not stable and varied from 13 to 31 g/L. The infiltrate was a multicomponent alkaline solution with pH 9.5–11.0. Its major components were chlorides, sulfates, carbonates, bicarbonates, nitrates, and ammonium ion at concentrations from hundreds of mg/L to a few g/L. The most common cations in solution were $Na^+$ (2–3 g/L), $Ca^{2+}$ (about 0.6 g/L), and $K^+$ (about 0.14 g/L). The distinctive feature of the composition of the waste solution was the absence of magnesium. The foundation soil massif is subdivided into two sections. The first is the zone of aeration that lies within the alluvial sands and sandy loams. The second one is the zone of saturation, which coincides with the weathered crust of the Jurassic sandstones.

The average concentration of uranium in the solid part of the sludge is 240 ppm; the total estimated amount of uranium in the repository is about 22 tons. The Baikal region is generally characterized by a higher content of uranium in different soils than other regions of Siberia (Russia). The average content of uranium is 1.66 ppm in the bedrock and 2.4–3.6 ppm in the soil [16]. Obviously, the high background content of uranium in rocks and soils is associated with the Irkutsk coal basin. There are heavy metals and radionuclides in accessory minerals, mainly, and in the clay fraction. We found that the content of these elements in the clay fraction is 2–3.5 times higher than in bulk samples. An even higher content was detected in a sample from a carbonaceous sublayer where the concentration of uranium was 30 times higher and reached 31.5 ppm [17].

Infiltration of a pollutant from storages results in the anomalous content of uranium and other elements in the soil and groundwater; it also leads to changes in the structure and mineralogical composition of the foundation soils. The maximum concentration of uranium in the soil under the two repositories is 5.6 and 11 ppm. It is quite important that a rather low concentration of uranium has been detected in the anomalies directly under the storage bottom. The concentration of uranium in the waters directly under the waste ponds is currently in some cases 5–9 times higher than the background, but at a distance of the first hundreds of meters, it decreases to the background level of 0.2–1 μg/L [17]. However, the long-term influence of waste brines on the adjacent rocks resulted in the modification of their mineralogical composition. Waters saturated with erosion products of cement slurries with wastes are called technogenic in this work.

To estimate the risk of uranium pollution outside the repository, we conducted research on the interaction between the adjacent soils and the infiltrating solutions and the effect of this interaction on the sorption and filtration properties of soils. In the zone of direct contact between the highly saline alkaline solutions and the foundation soil, considerable transformations were revealed.

## 3. Materials and Methods

Figure 1 shows wells (cyan) that were used for monitoring and planning further studies and exploratory wells (C1–C5, Figure 2), which were performed by auger drilling with sampling every 0.5 m, or when the parameters of the rocks changed. The depth of the wells varied from 8 to 12 m and was determined by the depth of the top of the sandstone layer. After extraction, the core samples were dried to an air-dry state. The objects of investigation are several samples collected from the contact zone between highly saline infiltrates under the waste storage and soils.

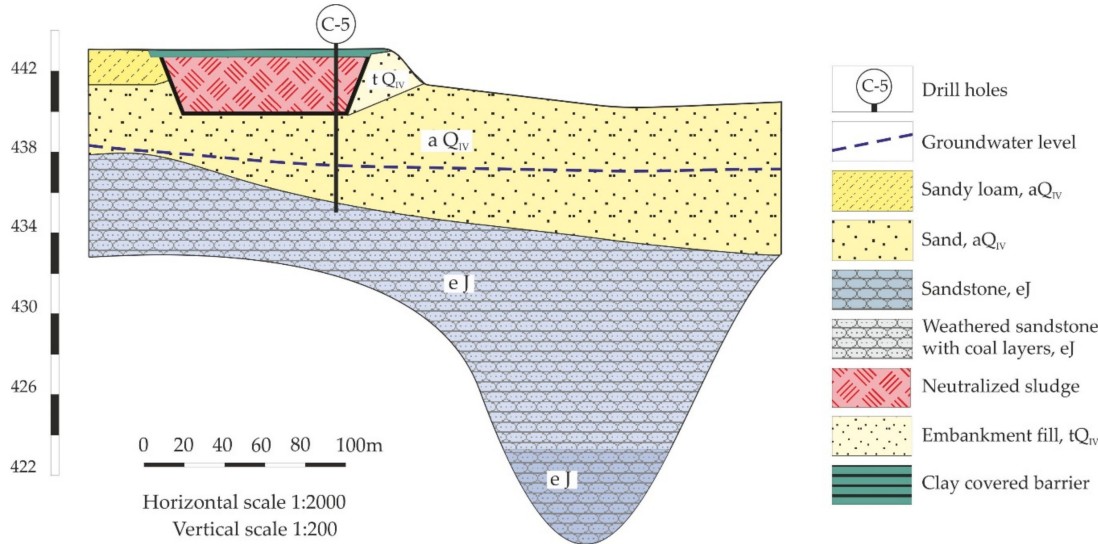

**Figure 2.** Cross-section through the reservoir (storage).

The mineral composition of the soils and its alteration were studied by the combination of different methods.

Mineral analysis was carried out by X-ray diffraction (XRD) for bulk samples and clay fractions (<1 μm). XRD patterns were obtained with an Ultima-IV X-ray diffractometer (Rigaku). The measurement conditions were Cu-Kα radiation, D/Tex-Ultra 1D-detector, and scan range of 3–65° 2θ. Identification of clay minerals was carried out on the XRD patterns from oriented mounts in the air-dried samples and ethylene glycol solvated. Non-oriented samples of fine fractions were used for determination of di-tri-octahedral species [18]. Quantitative analysis was carried out by the Rietveld method [19] with PROFEX GUI for BGMN [20] on the XRD patterns obtained from the random powder specimens of bulk samples after determination of the composition and structure of

clay minerals in the clay fraction. The accuracy of this approach to quantitative analysis is considered about 2–3% for each individual mineral phase.

For a more detailed identification of clay minerals, a fraction < 1 µm was separated by the standard sedimentation procedure according to Stokes' law. In order to avoid modification of clay minerals, no chemical treatment was carried out; to avoid coagulation, if necessary, the samples were repeatedly washed with distilled water and dispersed by ultrasonic treatment.

Chemical analysis of soils was carried out by X-ray fluorescence analysis using synchrotron radiation with a Si (Li) detector with the VEPP-3 elemental analysis station at the Institute of Nuclear Physics, Siberian Branch, Russian Academy of Sciences.

The content of natural radionuclides was determined by scintillation gamma spectrometry (SGS) at the center for collective use of the Institute of Geology and Mineralogy, Siberian Branch, Russian Academy of Sciences (IGM SB RAS). Contents of Ra (by decay products of 222 Rn), 232 Th, 40 K were determined by the intensity of gamma radiation using a low background well-type scintillation detector NaI(Tl). The weight of the sample was 200–300 g, and the detection limit of the method is 0.4 Bq/kg.

Micromorphology study was carried out with the scanning electron microscope (SEM) LEO 1450 VP (Carl Zeiss, Germany). Samples were air-dried and then fixed on the conductive tape. To eliminate the charging effect, they were coated with gold. The samples were studied in the secondary electrons' mode, with an accelerating voltage of 30 kV.

Infrared spectroscopy analysis was carried out using FTIR spectrometer Spectrum One (PerkinElmer) equipped by $LiTaO_3$ detector and KBr beam-splitter. The adsorption spectra recordings were performed in the 4000–400 $cm^{-1}$ wavelength range with 100 scans for each sample and the resolution of 4 $cm^{-1}$. Samples were prepared as pressed KBr-pellets: 1 mg of sample was dispersed in 400 mg of KBr; this mixture was placed in a 2 cm pellet die and pressed for 20 min. The KBr pellets were then placed into a glass desiccant box with $CaCl_2$ and heated overnight in a furnace at 150 °C. Spectra manipulations were performed using the OPUS 7.1 software (Bruker Optic GmbH, Ettlingen, Germany). Baseline correction was made by Straight-Line method with 1 iteration in interactive mode.

## 4. Results and Discussion

The adjacent soil layer is subdivided into two main layers: (1) quaternary alluvial sands with thin layers and lenses of sandy loams and loams and (2) weathered Jurassic sandstones with inclusions of carbonaceous matter.

The minimum concentration of the most microcomponents was noted in sands and sandstones. A lower content is typical for sandy soils consisting mainly of quartz and plagioclase (albite). Heavy metals and radionuclides are mainly associated with accessory minerals, and also with the clay fraction in the adsorbed state. The amount of this fraction changes from 13% to 28%. More than 30% is represented by clay minerals: kaolinite, smectite, and illite, which is confirmed by XRD analysis and is shown below. The content of most elements is 2–3.5 times higher in the clay fraction. The background elemental composition for the main types of soil in the studied area is presented in Table 1.

The water in the zone of saturation differs by lower salinity due to the dilution of highly saline infiltrates by natural groundwater [14]. In addition, this zone is also distinguished by the soil composition. The process of technogenic mineral alteration was superimposed on the naturally altered soils of the weathered crust. Transformation of rocks by highly saline waste infiltrates is accompanied by the removal of large amounts of cations, which is revealed by the changing of the qualitative chemical composition of groundwater: natural waters of the region have a cation concentration proportion Ca > Mg > Na, while waters below the storages are characterized by the proportion Na > Ca > Mg.

The content of calcium and magnesium increases due to the ion exchange and the dissolution of solid phases. Fine-crystalline phases may be accumulating in the pore space of soils; however, their presence has not been confirmed by XRD analysis data. The chemical composition shows no accumulation of uranium in the aeration zone. This is explained by the chemical composition of highly

saline solutions with a high content of nitrate ion, which determines the redox conditions [14,17,21] because in its presence uranium exists in the highly mobile form U(VI).

**Table 1.** Chemical composition of different types of soils.

| | Jurassic Sands C-3/11.8 m | Alluvial Quaternary Sands C-3/7.5 m | Low-altered Sands C-4/6.0 m | Intermediate-Altered Sands C-5/6.0 m | Highly Altered Sands C-5/7.5 m |
|---|---|---|---|---|---|
| K, % | 2.5 | 2.02 | 1.75 | 2.25 | 2.22 |
| Ca, % | 3.63 | 0.79 | 0.9 | 1.37 | 0.85 |
| Ti, % | 0.55 | 0.423 | 0.313 | 0.473 | 0.432 |
| Mn, % | 0.09 | 0.051 | 0.111 | 0.401 | 0.122 |
| Fe, % | 6.47 | 4.42 | 4.25 | 4.14 | 7.07 |
| V, ppm | 158 | 95 | 98 | 118 | 122 |
| Cr, ppm | 167 | 110 | 85 | 75 | 98 |
| Ni, ppm | 105 | 62 | 123 | 66 | 84 |
| Cu, ppm | 60 | 52.8 | 17.7 | 22 | 23.4 |
| Zn, ppm | 73 | 78 | 48.5 | 64 | 73 |
| Rb, ppm | 55 | 59 | 60 | 84 | 74 |
| Sr, ppm | 270 | 214 | 222 | 282 | 214 |
| Y, ppm | 19.9 | 13 | 13.4 | 21.5 | 14.8 |
| Zr, ppm | 170 | 88 | 156 | 120 | 102 |
| Nb, ppm | 7.01 | 5.59 | 5.03 | 9.02 | 9.48 |
| Mo, ppm | 0.36 | 0.7 | 1.19 | 1.07 | 1.41 |
| Pb, ppm | 13.8 | 14.2 | 17.4 | 15.7 | 15.8 |
| Th, ppm | 2.5 | 3.3 | 3.6 | 5.2 | 3.8 |
| U, ppm | >1 | >1 | 6.4 | 1.4 | >1 |

The mineral composition of the studied soils is quite typical for the studied area and is represented by quartz, K-feldspar, plagioclase, carbonates, and amphiboles—also as clay minerals: smectite, chlorite, illite, and kaolinite (Figure 3). XRD patterns show a decrease of smectite content and an increase in quartz content in alluvial sands compared to Jurassic sands (Table 2, estimation was made after detailed investigation of mineral composition of bulk and clay fraction that will be shown below).

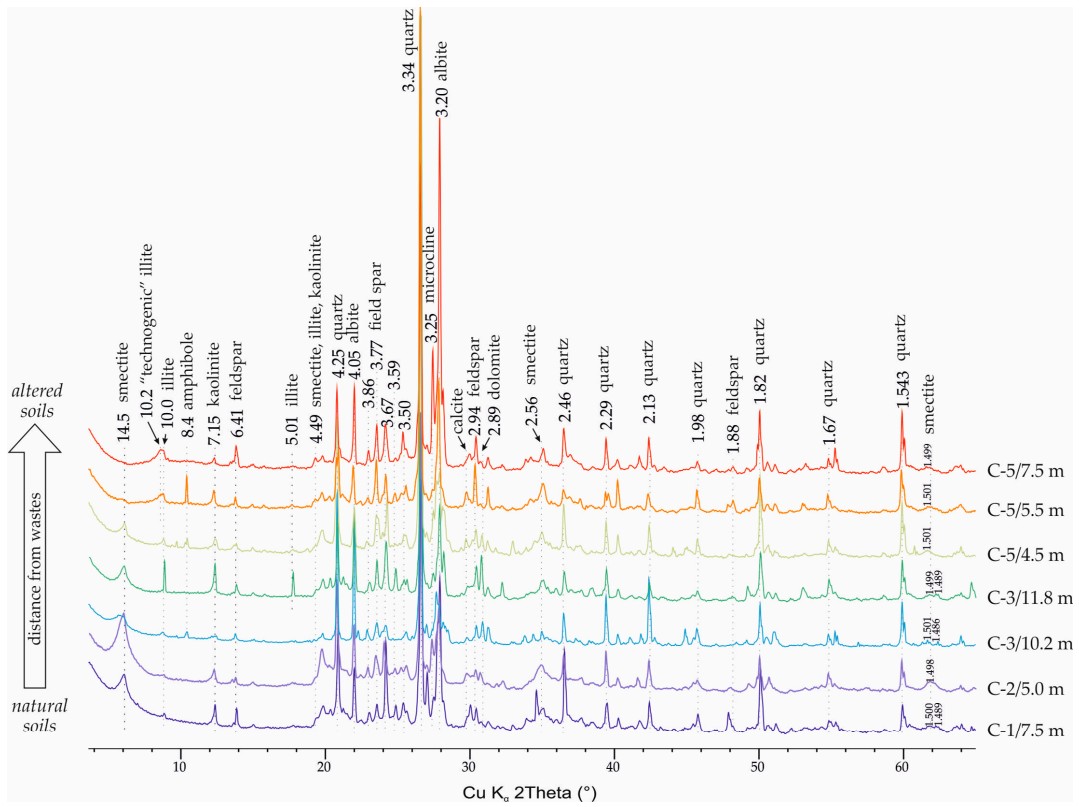

**Figure 3.** X-ray diffraction (XRD) patterns of the bulk samples. D-values are given in A.

**Table 2.** Mineral composition of the main types of soil.

| Mineral Composition | Jurassic Sands C-3/11.8 m | Alluvial Quaternary Sands C-3/7.5 m | Low-Altered Sands C-4/6.0 m | Intermediate-Altered Sands C-5/6.0 m | Highly Altered Sands C-5/7.5 m |
|---|---|---|---|---|---|
| smectite | 31.1 | 10.7 | 10.7 | 0.0 | 0.0 |
| illite and TI* | 4.8 | 5.8 | 13.5 | 28.7 | 33.9 |
| kaolinite | 4.6 | 5.1 | 1.1 | 5.7 | 3.0 |
| chlorite | 4.8 | 3.0 | 5.6 | 4.1 | 3.9 |
| carbonates | 0.6 | 4.8 | 4.5 | 0.8 | 0.9 |
| quartz | 15.4 | 32.2 | 29.5 | 26.3 | 15.3 |
| microcline | 14.3 | 10.7 | 8.8 | 8.6 | 11.4 |
| albite | 23.0 | 25.2 | 23.5 | 23.1 | 30.3 |
| amphibole | 1.4 | 2.5 | 2.7 | 2.8 | 1.3 |

TI*: "technogenic illite", explanation in the text below.

There is a clear change in the diffraction patterns of the transformed soils, in the range of 3–15°2θ. These changes are observed quite clearly in the samples of the lower horizons of well C5 (5.5–7.5 m), which was drilled directly through the maps and sedimentary rocks, which are affected by technogenic waters.

For more confident identification of technogenic changes in soils, the clay fraction of soils was analyzed. XRD patterns from oriented mounts and fragments of patterns from non-oriented mounts in the region of (060) peaks are shown in Figure 4. The presence of di- and tri-smectite varieties was noted in all the studied samples. At the same time, the peculiarities in XRD patterns from oriented specimens suggest that the swelling component is represented not only by di- and tri-smectites but also by mixed-layer minerals of the chlorite–smectite series with a predominance of smectite interlayers, which requires further, more detailed studies.

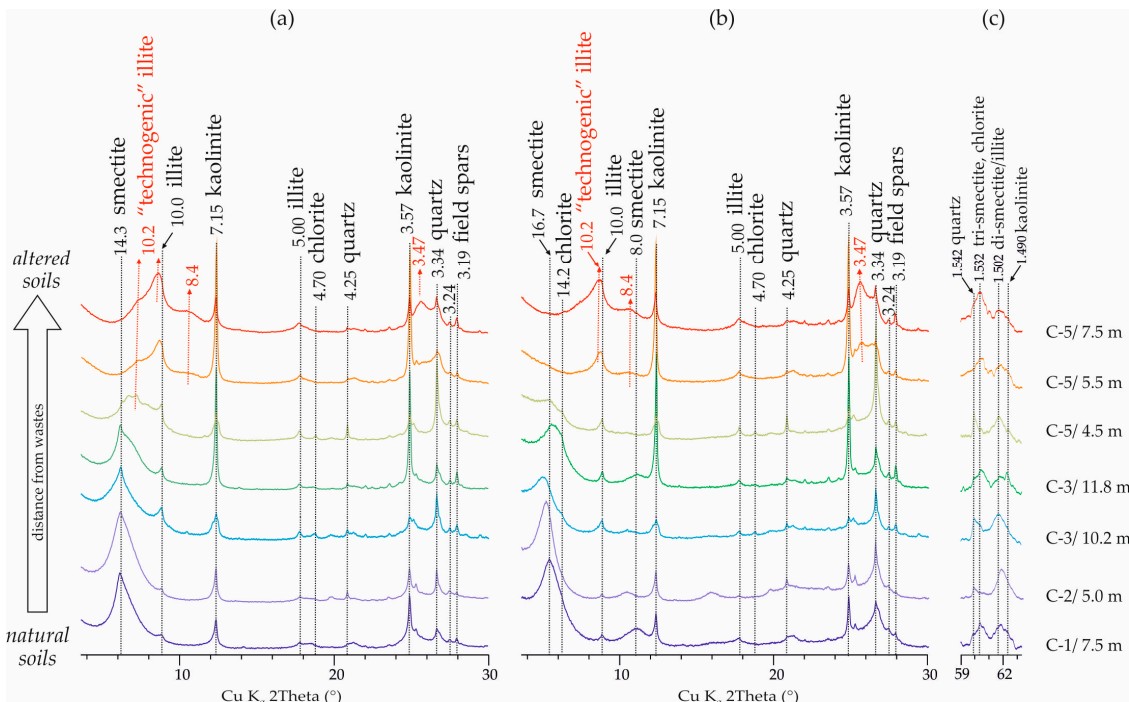

**Figure 4.** Fragments of the XRD patterns of oriented specimens (**a**,**b**) and non-oriented specimens (**c**). a—air-dried state, b—ethylene glycol solvated, c—(060) area. D-values are given in angstroms. TI peaks are marked by red color.

XRD patterns in the (060) part indicate that the visible changes in the composition of di- and tri-octahedral clay minerals in the transformed sands in comparison with the background soils may be caused by changes in the composition of the original soils and cannot be attributed to technogenic

changes. At the same time, changes in the XRD patterns from clay fractions (Figure 4) in the small-angle region are obvious and can be observed not only in the deeper horizons (5.5–7.5 m) of the well 5, as noted above, but also in the horizon 4.5 m of the well C5 and in the lower horizon (11.8) of the well C3. Samples C1/7.5, C3/5.0, and C3/10.2 can be classified as sands that have not undergone visible alterations due to the impact of industrial waters.

Changes in samples C3/11.8 and C5/4.5 are expressed by the appearance of peaks at 12.2 and 10.2 Å. In samples C/5.5 and C/7.5, the peak at 12.2 Å disappears, while the 10.2 Å remains and becomes more intensive; also, a peak 8.4 Å appears and both of them do not shift in XRD patterns from ethylene glycol solvated specimens.

The infra-red (IR) spectroscopy data of the fine fractions of non-altered C-3/11.8 and altered C-5/7.5 soil samples show the polymineral composition: predominantly dioctahedral smectite, chlorite (or mixed-layer minerals with chlorite layers), and kaolinite. Identification of minerals was performed in accordance with recommendations of [22,23]. Wavenumber values and profiles of the absorption bands in the IR spectra (Figure 5) for both samples are similar; the noticeable difference is caused only by a higher content of kaolinite in the altered soil specimen. The IR-spectroscopy data show that there are no "illitic" phases in the altered sample, otherwise, there would be a small band at $\approx 420$ cm$^{-1}$ on the spectra [24].

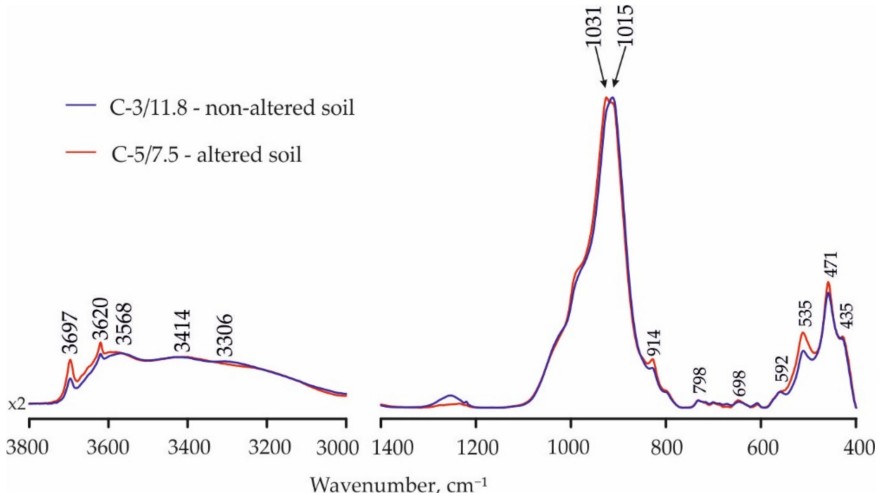

**Figure 5.** Fragments of the IR-spectra of non-altered and altered soil samples.

Electron microscopy reveals that the host rock was transformed significantly. Leached feldspars and neoformed (altered) clay minerals with an "openwork" structure can be observed (Figure 6c,d). The relict "openwork" structure of the technogenic illite (TI) indicates a rather fast transformation of smectite into illite without change of micromorphology and interaction between particles.

Among other clay minerals, the formation of chlorite on the surface of plagioclase grains was observed. At a depth of about 5 m under the storage bottom where the pH is close to neutral, neoformed opals were found (Figure 6e,f). We assume that the observed processes take place in the soils with a lack of silica caused by weathering, thus, the authigenic opal was formed as a result of a decrease of pH and precipitation of $SiO_2$ from the supersaturated solution.

Microstructures observed with a scanning electronic microscope are quite typical for weathered sandstones. Quite large isometric particles and aggregates of dense clay particles (probably kaolinite, illite, and smectite) and thin openwork smectite domains are clearly visible (Figure 6 a,b).

Thus, in samples of soil that were subjected to the filtration of technogenic waters, a specific clay phase is noted. It is most likely di-octahedral, which, according to IR spectroscopy and scanning electron microscopy data, is close to smectite but, according to XRD data, loses the ability to swelling and behaves more like illite, while not being illite exactly. Since this phase was found in technogenically

altered soils, it was named "technogenic illite"—TI, as previously suggested [25]. The loss of the swelling ability can significantly affect the insulating properties of soils, therefore, the phase was assigned to "illite" and not to "smectite", like the "technogenic smectites" previously described by the authors in the sands-collectors of liquid radioactive waste at the facility of Siberian Chemical Combine [12].

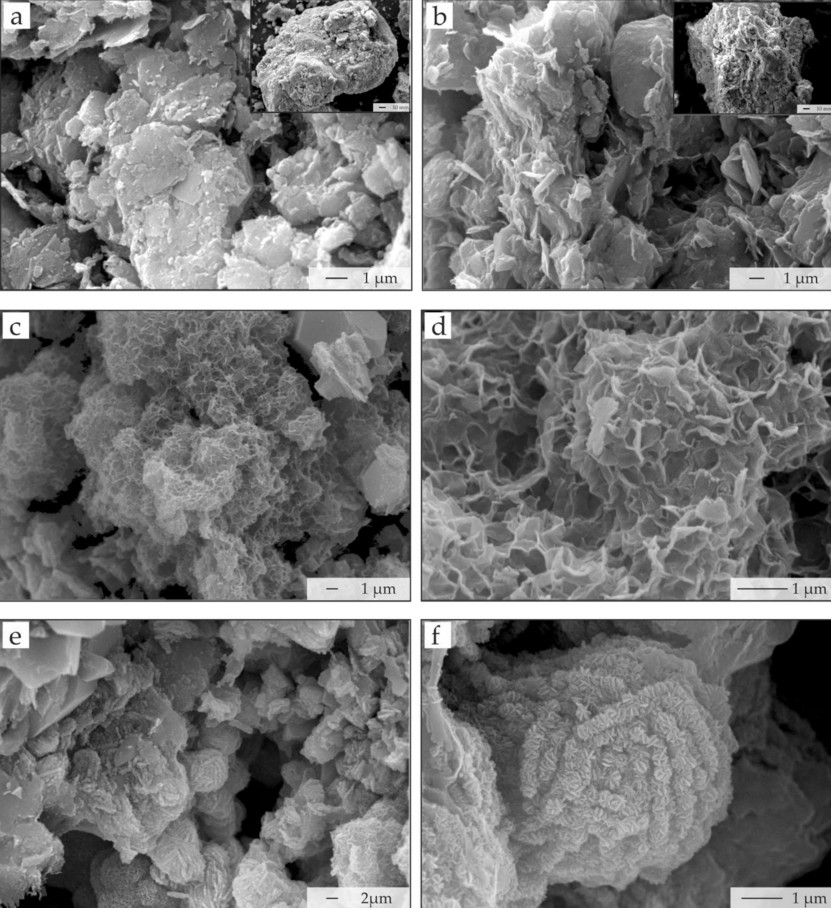

**Figure 6.** The microstructure of soils from different depths. Background soils: (**a**) loam from the depth of 7.5 m (Core C-3, aQIV); (**b**) weathered sandstone from the depth of 11.8 m (Core C-3 eJ). Underlying soils: (**c**) and (**d**) weathered sandstone from the depth of 7.5 m (Core C-5, aQIV); (**e**) and (**f**) neoformed opal from the depth of 7.5 m (Core C-5, aQIV).

Since the structural parameters of the new phase are unknown at the moment, a modified illite phase with modified parameters of the unit cell was incorporated into the model for calculating the quantitative content. The mineral composition is represented in the Table. 2, from which it is possible to trace the gradual increase of the TI content in weakly transformed soils and its predominance in technologically modified soils.

Based on the research results, it is possible to recommend conducting detailed studies of the composition of soils and, in particular, the composition of clay minerals when monitoring nuclear and radiation hazardous facilities and when implementing measures for their decommissioning.

## 5. Conclusions

Significant changes of the soils in the area of a long-term impact of the solutions from the low-radioactive liquid waste storages were revealed in the mineral composition of soils under the influence of waste infiltrates compared to the background samples. The transformation of minerals

especially clay minerals and the appearance of new mineral phases causes a change of the soil properties: permeability and sorption capacity, mainly.

Thus, directly under the bottom of storage throughout about two meters to groundwater level, there are no favorable conditions for the formation of a geochemical barrier that could prevent the migration of uranium. Although, below the groundwater level, the hydrochemical environment changes, which significantly reduces the effect of nitrate ion on the mobility of uranium, it also could decrease the sorption capacity of the soil at the observed area to a depth of about 3 m.

In the zone of influence of technogenic waters, "technogenic illite" was found. Many of its properties are similar to dioctahedral smectite, however, it does not swell, which primarily distinguishes smectites from illites. The loss of the swelling capacity of "technogenic illites" in the clay fraction of soils can lead to a decrease in the waterproofing properties of rocks and potentially increase the risk of pollution by technogenic waters.

**Author Contributions:** Conceptualization, V.K. and A.B.; data curation, V.K. and S.Z., methodology, V.K. and S.Z.; formal analysis, S.Z., O.S., M.C., O.D., and I.M.; investigation, V.K., S.Z., M.C., and O.D.; writing—original draft preparation, V.K., A.B., O.S., and S.Z.; writing—review and editing, V.K., A.B., O.S., O.D., and S.Z. All authors have read and agreed to the published version of the manuscript.

**Funding:** This research was done on state assignment of IGM SB RAS. Investigations of clay minerals transformation were carried out within the framework of the project IGEM RAS AAAA-A18-118021590167-1. Experimental studies were partially performed on the equipment acquired with the funding of the Moscow State University Development Program (X-ray diffractometer Ultima-IV, Rigaku and scanning electron microscope LEO 1450VP, Carl Zeiss).

**Acknowledgments:** The authors are grateful to Svetlana Garanina (Lomonosov Moscow State University) for assistance in XRD measurements.

**Conflicts of Interest:** The authors declare no conflict of interest.

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
