# Peer review of "The Influence of Liquid Low-Radioactive Waste Repositories on the Mineral Composition of Surrounding Soils"

_sustainability, doi:10.3390/su12198259_

Round 1

Reviewer 1 Report

In this manuscript the authors investigate the influence of alkaline solutions involved in the radioactive waste (RW) disposal process. Solid radioactive wastes in cement slurries at the landfill site of the Angara Electrolysis Chemical Combine are buried in sandstones and currently suffer the influence of highly alkaline and highly saline groundwater storage on sandstones and soils at the landfill site of the Angara Electrolysis Chemical Combine. One of the consequences is the formation of what are called technogenic illites that have smectite morphology but illite structure. This research is focused on the question regarding the fate of radioactive waste repositories. A key problem facing the decommissioned facilities is whether the existing barriers are efficient enough to isolate radioactive wastes. The underlying and host soils have low permeability together with high sorption characteristics, and therefore radioactive waste recovery and retrieval can carry serious environmental risks compared with the in situ preservation. Following detailed mineralogical, geochemical and scanning electron microscope studies the authors conclude that significant changes of the soils in the area subjected to a long-term impact of the solutions from the low radioactive liquid waste storages were revealed in the mineral composition of waste infiltrate soils compared to the background samples. This is important information, not only to the clay science community but to the larger area of environmental geoscience. The authors have done an excellent job of collecting, analyzing and describing their samples and this report is very appropriate for publication in Sustainability. My only suggestion is to offer the authors several English grammatical changes that they might find helpful. They are as follows:

Line 25: …..storage area, which leads to…..

Line 50: …..impact assessment and a study of the forms of migration and the state of barriers….

Line 55: …..systems under geological,…..

Line 56: …..processes are believed to be the most sensitive towards the host geological…..

Line 78: …..scheme of the AECC was intended to cause precipitation…..

Line 87: …..to the research from…..

Line 88: …..the salinity of waste…..

Line 91: …..(about 0.6 g/L), and K+…..

Line 96: …..Obviously, the high background…..

Line 97: …..and in the clay fraction.

Line 98: …..We found that the content………..An even higher….

Line 101: …we conducted research on…..

Line 111: …The mineral composition of the soils…..

Line 127: …scintillation detector…………200-300 g, and the detection…..

Line 140: …The weathered sandstone layer…..

Line 142: …The clay fraction content…..

Line 150: …Mineral composition is represented in Table 1.

Line 158: …A lower content is typical…..

Line 161: …The content of most elements…..

Line 163: …presented in Table 2.

Line 167: …observed with a scanning electron microscope…..

Line 182: …zone. On the contrary…..

Line 234: …The infiltration of highly saline…..

Author Response

Authors are very grateful for such a high evaluation of the conducted research and also for the Reviewer’s effort for improving of the article. All the comments and suggestions were revised and the corrections have been made. 

Line 25: …..storage area, which leads to…..

Corrected, thank you very much

Line 50: …..impact assessment and a study of the forms of migration and the state of barriers….

Corrected, thank you very much

Line 55: …..systems under geological,…..

Corrected, thank you very much

Line 56: …..processes are believed to be the most sensitive towards the host geological…..

Corrected, thank you very much

Line 78: …..scheme of the AECC was intended to cause precipitation…..

Corrected, thank you very much

Line 87: …..to the research from…..

Corrected, thank you very much

Line 88: …..the salinity of waste…..

Corrected, thank you very much

Line 91: …..(about 0.6 g/L), and K+…..

Corrected, thank you very much

Line 96: …..Obviously, the high background…..

Corrected, thank you very much

Line 97: …..and in the clay fraction.

Corrected, thank you very much

Line 98: …..We found that the content………..An even higher….

Corrected, thank you very much

Line 101: …we conducted research on…..

Corrected, thank you very much

Line 111: …The mineral composition of the soils…..

Corrected, thank you very much

Line 127: …scintillation detector…………200-300 g, and the detection…..

Corrected, thank you very much

Line 140: …The weathered sandstone layer…..

Corrected, thank you very much

Line 142: …The clay fraction content…..

Corrected, thank you very much

Line 150: …Mineral composition is represented in Table 1.

Corrected, thank you very much

Line 158: …A lower content is typical…..

Corrected, thank you very much

Line 161: …The content of most elements…..

Corrected, thank you very much

Line 163: …presented in Table 2.

Corrected, thank you very much

Line 167: …observed with a scanning electron microscope…..

Corrected, thank you very much

Line 182: …zone. On the contrary…..

Corrected, thank you very much

Line 234: …The infiltration of highly saline…..

Corrected, thank you very much

Reviewer 2 Report

The manuscript deals with a highly important issue of soil and groundwater contamination around the radioactive waste repositories. The authors investigated chemical and mineralogical composition of several samples collected in the vicinity of sedimentation ponds storing waste water from an uranium recovery facility in Irkutsk Region, Russia. Based on the obtained results, the authors report mineralogical alteration sequence caused by the seepage of the highly alkaline brine from the ponds. The manuscript contains valuable results that will surely be of high interest to the scientific community. Nevertheless, there are several aspects of the work which caused my concern. I would like to ask the authors to address the following issues:

  1. I would recommend to improve sections 1 and 2. The authors give clear background on the nuclear waste repositories in Russia but very little information is given about the current state of knowledge about mineralogical transformation of soils near nuclear waste repositories similar to the one investigated by the authors. Were similar research conducted? If yes, what were their main findings? Also, the authors use term “technogenic” illite without much introduction. I suggest adding some information about “technogenic illite” in the introduction and explaining, what is the difference between “technogenic” and “non-technogenic” illite.

Section 2 (the first one, “Geological position…”) is generally clear and well written, but I would suggest to include in it all the information about the geology and hydrogeology of the studied site, which are now scattered across the discussion (lines 177-179, 184-186).

  1. The authors should a little more details about their clay fraction separation procedure. Were there any chemical pretreatments involved, such as the Jackson procedure? This is important and should be stated explicitly.
  2. The authors show XRD diffraction patters on oriented preparations only. I suggest to add examples of disoriented preparations with fits obtained in PROFEX-BGMN software as well. Presenting this data will increase credibility of results presented in Table 1 and will support identification of smectite present in the samples as dioctahedral variety. Also, because the authors compare composition of bulk material and clay fractions, I suggest showing and comparing clearly results for both bulk material and clay fractions. This applies to both XRD and chemical results.
  3. I would recommend to give more information about U concentration in the investigated samples. This is a vital aspect of the work. I suggest adding a systematic description of how U concentration changes with depth and distance from the ponds.
  4. There are some minor editorial issues with the text. There are two sections 2 (“Geological position…” and “Materials and Methods”). Please change number of the “Material and Methods” section to 3 and correct the following sections’ numbers accordingly. Also, numbers of positions 3 to 8 in the reference list are doubled. Please remove redundant reference numbers.

Line-by-line comments

Line 25 – I suggest removing “described above”. It is redundant.

Line 27 – In the present form, there are no XRD models presented in the manuscript that supports this conclusion. The authors mention, albeit not show (see general comments) modeling of disoriented preparations. Figure 4, in turn, presents no models. I suggest expanding sections on XRD patterns modeling in order to support this claim or to change this sentence, because it is a little misleading.

Line 38-39 – Please add a reference to the description of the federal decommissioning program, if possible.

Line 67 – “U” should not be in the superscript. Should read “235U”.

Line 84 – The information about fluid migration from the ponds to the adjacent soil/rocks is very important. It is a little unclear if this is your observation or this comes from earlier research. Please add a reference, if the latter is true.

Line 87 – The citation format of the Matveeva et al. (2009) is not consistent with other references in the text. Please, use citation number instead of first author’s name.

Line 94 – It is unclear what the authors compare uranium content of the Baikal region to. The content is higher than what? Did you mean the average uranium content in the other parts of the Russian Federation? Please, clarify. The same applies to line 98 – Content of the heavy metals and radionuclides in clay fractions is 2-3.5 times higher than what?

Line 108-109 – The sentence starting with “All the observed changes…” is a conclusion. Please move it farther in the text.

Line 137 – Why reference to Figure 2 here? Figure 2 is a cross section through the repository and it does not contain any mineralogical data. Did you mean Table 1?

Line 141 - How do you know that smectite is montmorillonite? This is expected, but it would be good to give evidence about dioctahedral nature of the swelling clays. In addition, Green-Kelly test is required to differentiate between montmorillonite and beidellite. Because it was not conducted, it would be safer to address this phase as dioctahedral smectite here and throughout the manuscript.

Line 143 – The clay content is high in which samples exactly? Please add a reference to a Table or Figure.

Line 146 – Please change “accounted” to “accounted for” if this keeps your meaning.

Line 150 – The sentence starting with “Mineral composition…” is redundant and should be removed. I suggest adding references to Table 1 earlier in this paragraph, were you refer to data presented in it.

Lines 157-163 - This paragraph should be reorganized. Chemical and mineralogical information is mixed together, which makes it hard to follow. Line 160 - 30% of what? Clay fraction? Also, Table 2 shows composition only of bulk materials (presumably), while the authors compare compositions of bulk soils and clay fractions here. Please show mineral composition of both bulk material and clay fractions if you want to compare them.

Lines 180 – 181 - XRD is not capable of detection phases in the pore space. It gives you the bulk composition of a material. I assume that you detected presence of gypsum and other salts by XRD and the rock texture (i.e. the fact that gypsum and salts are in the pore space) with other method (SEM?). Please, clarify.

Lines 189-190 - This is a conclusion. Please, give evidence for that first.

Line 207 - What do you mean by a “modified interlayer complex”? It seems that interlayer cations were exchanged from Ca-Mg to Na, probably due to the laboratory treatments. However, the authors should control composition of the exchangeable complex in their samples.

Line 221 – A space is missing between “5” and “m”. Should read “5 m” The same applied to the line 244. More importantly – How did you measure pH of the pore fluid? Was it measured in situ in the field? Please, provide description or reference.

Figure 1 – The authors show positions of a numerous drill holes in the figure, but only results for the drill holes C1 to C5 are presented in the text. I would suggest removing the boreholes which you did not investigate in order to clarify the figure. Now it is unnecessarily busy.

Table 2 – I would suggest adding descriptions of samples for which data are presented, similar to Table 1.

Author Response

Dear Rewiever 2!

The authors are extremely grateful to the Reviewer for the thorough work with the materials and extremely useful questions, comments and recommendations, which, we hope, made it possible to make the article better.

Below are the answers points by points

  1. I would recommend to improve sections 1 and 2. The authors give clear background on the nuclear waste repositories in Russia but very little information is given about the current state of knowledge about mineralogical transformation of soils near nuclear waste repositories similar to the one investigated by the authors. Were similar research conducted? If yes, what were their main findings? Also, the authors use term “technogenic” illite without much introduction. I suggest adding some information about “technogenic illite” in the introduction and explaining, what is the difference between “technogenic” and “non-technogenic” illite.

The recommendation was very useful. We have revised the Introduction to add the citations on the known researches on soil transformation. Unfortunately, studies devoted to changes in the mineral composition of soils in the vicinity of radioactive waste disposal sites are rather rare. If the reviewer knows such works that are not mentioned in the manuscript and would not mind sharing them, then the authors will be extremely grateful!

The remark about the introduction of the term "technogenic illite" is very correct. Indeed, the introduction of the term at the very beginning of the Chapter "Results" without additional explanation does not seem clear enough.

The order of the manuscript was completely redone, explanations were added, the text was rewritten.

Section 2 (the first one, “Geological position…”) is generally clear and well written, but I would suggest to include in it all the information about the geology and hydrogeology of the studied site, which are now scattered across the discussion (lines 177-179, 184-186).

This study is focused not on hydrogeology, but on the changes in mineral composition, therefore, new data was not added. The issues of uranium migration are covered in detail in the paper:

Boguslavskiy, A.E., Gaskova, O.L., Shemelina, O.V., 2012. Uranium Migration in the Ground Water of the Region of Sludge Dumps of the Angarsk Electrolysis Chemical Combine. Chemistry for Sustainable Development 20, 465-478

  1. The authors should a little more details about their clay fraction separation procedure. Were there any chemical pretreatments involved, such as the Jackson procedure? This is important and should be stated explicitly.

Thanks to the Reviewer for a useful remark. Indeed, the use of different methods of sample preparation and separation of fractions can affect the results and change the XRD pattern. Explanations have been added to the text in the section "Materials and Methods".

For a more detailed identification of clay minerals, a fraction <1 μm was separated by the standard sedimentation procedure according to Stokes' law. In order to avoid modification of clay minerals, no chemical treatment was carried out; to avoid coagulation, if necessary, the samples were repeatedly washed with distilled water and dispersed by ultrasonic treatment.

  1. The authors show XRD diffraction patters on oriented preparations only. I suggest to add examples of disoriented preparations with fits obtained in PROFEX-BGMN software as well. Presenting this data will increase credibility of results presented in Table 1 and will support identification of smectite present in the samples as dioctahedral variety. Also, because the authors compare composition of bulk material and clay fractions, I suggest showing and comparing clearly results for both bulk material and clay fractions. This applies to both XRD and chemical results.

Of course, XRD patterns from the bulk samples provide a lot of useful information. The authors apologize for not including this material in the manuscript due to lack of time. The new Figure has been added.

  1. I would recommend to give more information about U concentration in the investigated samples. This is a vital aspect of the work. I suggest adding a systematic description of how U concentration changes with depth and distance from the ponds.

The authors are grateful for the question.

The regularities of the uranium behavior in groundwater are described in more detail in (Boguslavskiy, Gaskova, 2012). Therefore, only one phrase has been added to this article:

The concentration of uranium in the waters directly under the waste ponds is currently in some cases 5-9 times higher than the background, but at a distance of the first hundreds of meters it decreases to the background level of 0.2-1 μg/L

  1. There are some minor editorial issues with the text. There are two sections 2 (“Geological position…” and “Materials and Methods”). Please change number of the “Material and Methods” section to 3 and correct the following sections’ numbers accordingly. Also, numbers of positions 3 to 8 in the reference list are doubled. Please remove redundant reference numbers.

The authors are very grateful for such a thorough work with the manuscript! On my own behalf and on behalf of the co-authors, I would like express my gratitude to the reviewer once again.

Line-by-line comments

Line 25 – I suggest removing “described above”. It is redundant.

Deleted

Line 27 – In the present form, there are no XRD models presented in the manuscript that supports this conclusion. The authors mention, albeit not show (see general comments) modeling of disoriented preparations. Figure 4, in turn, presents no models. I suggest expanding sections on XRD patterns modeling in order to support this claim or to change this sentence, because it is a little misleading.

The XRD part of the manuscript has been significantly extended. Figure with XRD patterns of non-oriented specimens has been added. However, we decided not to give additional information on XRD modeling, because first of all this would lead to an increase in the number of figures and it seems that we would need new core samples and additional researches to investigate the peculiarities of the mixed-layer phases that we found.

Line 38-39 – Please add a reference to the description of the federal decommissioning program, if possible.

Added some reference and web-page

Line 67 – “U” should not be in the superscript. Should read “235U”.

Corrected

Line 84 – The information about fluid migration from the ponds to the adjacent soil/rocks is very important. It is a little unclear if this is your observation or this comes from earlier research. Please add a reference, if the latter is true.

The following explanation has been added.

As a result, a significant change in the groundwater composition is observed around the storage facilities.

The first aquifer is represented by the waters of Quaternary sediments, and the most saturated within the terraces above the floodplain. Depending on the topography, the aquifer lies at depths from 0.5 to 7.0 m. The water of the Quaternary aquifer is bicarbonate Ca>Mg>Na. Total mineralization ranges from 0.15 to 0.3 g/L. In the sampled area, the top of the technogenically altered groundwater flow occurs at a depth of 2.5 to 6.3 m. The groundwater from the SRW construction site moves in the north-north-east direction towards the Angara river, which is about 5.5 km in the north-east direction. Technogenic waters have a carbonate-nitrate composition (Na>Ca>Mg). Directly under the waste ponds, salinity in some wells reaches 9 g/L (C70), but already at a distance of 300 - 500 m (C78, 79) it drops to 0.3 - 0.5 g/L [17].

Line 87 – The citation format of the Matveeva et al. (2009) is not consistent with other references in the text. Please, use citation number instead of first author’s name.

Corrected

Line 94 – It is unclear what the authors compare uranium content of the Baikal region to. The content is higher than what? Did you mean the average uranium content in the other parts of the Russian Federation? Please, clarify. The same applies to line 98 – Content of the heavy metals and radionuclides in clay fractions is 2-3.5 times higher than what?

Thank you for the remark. Corrected.

Line 108-109 – The sentence starting with “All the observed changes…” is a conclusion. Please move it farther in the text.

Corrected

Line 137 – Why reference to Figure 2 here? Figure 2 is a cross section through the repository and it does not contain any mineralogical data. Did you mean Table 1?

Revised.

Line 141 - How do you know that smectite is montmorillonite? This is expected, but it would be good to give evidence about dioctahedral nature of the swelling clays. In addition, Green-Kelly test is required to differentiate between montmorillonite and beidellite. Because it was not conducted, it would be safer to address this phase as dioctahedral smectite here and throughout the manuscript.

The authors are very grateful for these comments and questions.

Indeed, among the swelling minerals, after a more careful analysis di- and tri-smectites were identified, as well as, possibly, chlorite (tri) -smectite (tri) mixed-layer phase. Unfortunately, the Green-Kelly test with such a mixture would not help. We carried out additional analysis by infrared spectroscopy and there is also no definite answer about the composition of mixed-layer minerals. Tri- and di- layered aluminosilicates are clearly separated in the (060) region. However, simulations have been carried out in Sybilla, but the results so far are not yet satisfactory enough to be published. The authors hope that new results will be obtained soon and can be included in the article.

Line 143 – The clay content is high in which samples exactly? Please add a reference to a Table or Figure.

Rewritten.

Line 146 – Please change “accounted” to “accounted for” if this keeps your meaning.

Thanks for the proofreading. The text was rewritten and this phrase was deleted

Line 150 – The sentence starting with “Mineral composition…” is redundant and should be removed. I suggest adding references to Table 1 earlier in this paragraph, were you refer to data presented in it.

Thanks for the proofreading. The text was rewritten and this phrase was deleted

Lines 157-163 - This paragraph should be reorganized. Chemical and mineralogical information is mixed together, which makes it hard to follow. Line 160 - 30% of what? Clay fraction? Also, Table 2 shows composition only of bulk materials (presumably), while the authors compare compositions of bulk soils and clay fractions here. Please show mineral composition of both bulk material and clay fractions if you want to compare them.

This part has been rewritten.

Lines 180 – 181 - XRD is not capable of detection phases in the pore space. It gives you the bulk composition of a material. I assume that you detected presence of gypsum and other salts by XRD and the rock texture (i.e. the fact that gypsum and salts are in the pore space) with other method (SEM?). Please, clarify.

Thanks for the proofreading. Rewritten.

Lines 189-190 - This is a conclusion. Please, give evidence for that first.

Thanks for the remark. Rewritten.

Line 207 - What do you mean by a “modified interlayer complex”? It seems that interlayer cations were exchanged from Ca-Mg to Na, probably due to the laboratory treatments. However, the authors should control composition of the exchangeable complex in their samples.

Thanks for the comment, this part has been deleted.

Changes in the composition of the exchangeable complex during laboratory treatments could not occur, since no chemical treatment was used in separation of clay fractions.

Line 221 – A space is missing between “5” and “m”. Should read “5 m” The same applied to the line 244. More importantly – How did you measure pH of the pore fluid? Was it measured in situ in the field? Please, provide description or reference.

Corrected. pH was measured in the pore fluid. Description are published before [14, 17]. It is not the main topic of the paper and would be added to the text.

Figure 1 – The authors show positions of a numerous drill holes in the figure, but only results for the drill holes C1 to C5 are presented in the text. I would suggest removing the boreholes which you did not investigate in order to clarify the figure. Now it is unnecessarily busy.

The figure has been corrected, corrections and additions have been made to the manuscript.

Table 2 – I would suggest adding descriptions of samples for which data are presented, similar to Table 1.

The table has been corrected, thanks for the comment

Thank you again for your hard work with our manuscript and hope that we can answer for the most part of you questions and comments.

Sincerely,

Victoria and co-authors

Reviewer 3 Report

  1. There are some points that the authors should correct “clay” with “clay minerals”, e.g. Abstract line 18, Conclusions line 233.
  2. Figure 2 and especially the explanation on the right of the Cross section should be corrected. They do not much and are not in the correct colour.
  3. Table 1. What is the error of the analyses?
Added Comments:

1. There are some points in the manuscript that the authors should correct
the term “clay” (which means the rock) with the term “clay minerals”
(which means the minerals), e.g. Abstract line 18, Conclusions line 233. 2. Figure 2 should be corrected because it is not clear what kind of rocks
are the rocks in the cross section. For example I cannot discriminate
where is the clay layer. I suggest that the authors should correct the
explanation on the right of the Cross section in order to be in accordance
and in the correct colour with the Cross section. 3. Table 1. What is the error of the semiquantitative analyses?
The authors present values like 31.1 %, 1,4%. If the accuracy is
about 2% or 3% the authors is better to write 31% and 1% respectively.
If the accuracy is below 1% the authors should state it and explain exactly
how they achieve that.

Author Response

On behalf of the authors I express our gratitude to the Reviewer for such a high assessment of our article and for the time and effort to help us in improving it. All comments were considered and the necessary corrections were made to the manuscript. Below are the responses to the comments.

Comments and Suggestions for Authors

  1. There are some points that the authors should correct “clay” with “clay minerals”, e.g. Abstract line 18, Conclusions line 233.

Thank you very much for the comment! The reviewer is right, it is more correct to use the term “clay minerals” Corresponding corrections have been made

Figure 2 and especially the explanation on the right of the Cross section should be corrected. They do not much and are not in the correct colour.

Thanks for the comments. The figure has been corrected.

  1. Table 1. What is the error of the analyses?

Reviewer asks a rather important question about the accuracy of the quantitative analysis method. We  consider that the accuracy is approximately 2-3% for each individual phase.

In fact we estimate the accuracy about 2-3% for individual phases, which gives an error of the order of hundredths or tenths of a percent, so the results have been rounded to tenths of a percent. Indicating this error value, we rely on what the Profex calculates (about 0.5-3% for individual phases) and on a several of works devoted to quantitative mineral analysis using XRD, including the well-known [J. Srodon et al., Quantitative X-ray diffraction analysis of clay-bearing rocks from random preparations. Clays and Clay Minerals. 49, 2001], and, for instance [L. León-Reina et al., Accuracy in Rietveld quantitative phase analysis: a comparative study of strictly monochromatic Mo and Cu radiations. J Appl Crystallogr. 2016 Jun 1; 49 (Pt 3): 722-735, 2016]

Answers for added Comments:

1. There are some points in the manuscript that the authors should correct 
the term “clay” (which means the rock) with the term “clay minerals” 
(which means the minerals), e.g. Abstract line 18, Conclusions line 233.

Thanks for the comments! Text has been corrected

  1. Figure 2 should be corrected because it is not clear what kind of rocks are the rocks in the cross section. For example I cannot discriminate where is the clay layer. I suggest that the authors should correct the  explanation on the right of the Cross section in order to be in accordance and in the correct colour with the Cross section.

Thanks for the comments. The figure has been corrected.

  1. Table 1. What is the error of the semiquantitative analyses? 
    The authors present values like 31.1 %, 1,4%. If the accuracy is 
    about 2% or 3% the authors is better to write 31% and 1% respectively. 
    If the accuracy is below 1% the authors should state it and explain exactly 
    how they achieve that. 

The accuracy of the method is important, thanks for the comment. Corresponding explanations have been added.

Sincerely,

Victoria Krupskaya and co-authors

Reviewer 4 Report

First of all, my congratulation to You on a very interesting topic of the manuscript.

The topic of the manuscript is soil, more precisely clay soil. In order for the behavior of clay soil to be clear to all readers, you must explain the behavior of clay soil in the introduction. You talk about the impact of liquid waste on the chemical composition of clay soil and changing the characteristics of clay, you need to support this with relevant factors. State the geotechnical characteristics of clay soil before and after the impact of liquid waste, and so on....
Figure 1 shows the drill borholes. In a separate chapter, explain and present the data of the drilled boreholes. How was the drilling performed, were the cores extracted, how was the data processed, what can be concluded from the obtained data, and so on ...

I suggest the following changes:

Line 77. - In Figure 1 mark the location of the sublimation unit and the separation unit. Make a legend. Correct the figure resolution and the text on the figure must be Palatino Linotype.
Line 87 - Fix references in the text according to the recommendations, for example ....ecological survey (Matveeva et al., 2009) must be ecological survey, Matveeva et al., 2009 [16], but take care of the reference number. Matveeva et al., 2009 is then [9]...and so on
Line 153 - Fix Resolution, add a legend and Font must be Palatino Linotype. The picture is in color and the legend in black and white. It is unclear where the clay layer is located.
Line 165 - Fix table 2. Table 2 should be on a new page.
Line 198, 227 - Fix Resolution and Font must be Palatino Linotype.
Line 258 - Fix references [3], [4], [5], [6], [7], [8]

Author Response

On behalf of the authors, thank you for your appreciation of the research topic and for your efforts to improve the article.

Bellow an answers and comments point by point.

Comments and Suggestions for Authors

First of all, my congratulation to You on a very interesting topic of the manuscript.

The topic of the manuscript is soil, more precisely clay soil. In order for the behavior of clay soil to be clear to all readers, you must explain the behavior of clay soil in the introduction. You talk about the impact of liquid waste on the chemical composition of clay soil and changing the characteristics of clay, you need to support this with relevant factors. State the geotechnical characteristics of clay soil before and after the impact of liquid waste, and so on....

The aim of the research was to study changes in the mineralogical and chemical characteristics of soils. The study of geotechnical parameters was not considered in this research.

The authors are grateful for this question and will attempt to include the geotechnical characteristics of soils in the continuation of research if we could obtain new undisturbed core samples.

Figure 1 shows the drill borholes. In a separate chapter, explain and present the data of the drilled boreholes. How was the drilling performed, were the cores extracted, how was the data processed, what can be concluded from the obtained data, and so on ...

The authors are grateful for the question. The necessary explanation should have been given in the manuscript, however, it turned out that we were much more on the laboratory part of the research at the expense of the field work. Samples were received in a disturbed state.

The following explanation has been added to the manuscript:

The figure 1 shows wells (cyan) that were used for monitoring and planning further studies and exploratory wells (C1-C5, fig.2), which were performed by auger drilling with sampling every 0.5 meters, or when the parameters of the rocks changed. The depth of the wells varied from 8 to 12 meters and was determined by the depth of the top of the sandstone layer. After extraction, the core samples were dried to an air-dry state.

I suggest the following changes:

Line 77. - In Figure 1 mark the location of the sublimation unit and the separation unit. Make a legend. Correct the figure resolution and the text on the figure must be Palatino Linotype.

The figure has been corrected

Line 87 - Fix references in the text according to the recommendations, for example ....ecological survey (Matveeva et al., 2009) must be ecological survey, Matveeva et al., 2009 [16], but take care of the reference number. Matveeva et al., 2009 is then [9]...and so on

Corrected.

Line 153 - Fix Resolution, add a legend and Font must be Palatino Linotype. The picture is in color and the legend in black and white. It is unclear where the clay layer is located.

Corrected.

Line 165 - Fix table 2. Table 2 should be on a new page.

Corrected.

Line 198, 227 - Fix Resolution and Font must be Palatino Linotype.

Corrected.

Line 258 - Fix references [3], [4], [5], [6], [7], [8]

Corrected.

Thank you very much for correction again!

Sincerely,

Victoria Krupskaya and co-authors

Round 2

Reviewer 4 Report

I suggest the following changes:

Line 219 - Moderate text, it is Table 2, and Table 2 must be on the same page